# Exploring the Impact of Social Identity on the Bullying of Construction Industry Apprentices

**DOI:** 10.3390/ijerph20216980

**Published:** 2023-10-26

**Authors:** Peter Greacen, Victoria Ross

**Affiliations:** Australian Institute for Suicide Research and Prevention (AISRAP), School of Applied Psychology, Griffith University, Brisbane, QLD 4122, Australia; peter.greacen@griffithuni.edu.au

**Keywords:** apprentices, construction industry, group membership, mental health, social identification, suicide, workplace bullying

## Abstract

Background: There is a lack of literature specifically examining the workplace bullying of apprentices and trainees in traditional, male-dominated sectors such as the Australian building and construction industry. Using social identity theory (SIT), the aim of this study was to gather the attitudes, thoughts, and feelings of construction industry leaders to better understand how social identification (i.e., group membership) impacts bullying on targets and perpetrators and the willingness to report bullying to targets and bystanders. Method: One-on-one, semi-structured interviews using a purposive sample of eight leaders from construction and blue-collar industries. Qualitative data were analysed using reflexive thematic analysis. Results: Four overarching themes were identified: difficulties for apprentices transitioning into industry, the need for continued improvement in industry culture, reluctance to report bullying, and rethinking apprenticeships to empower. Each theme provides insight into the psychosocial phenomenon of the bullying of trade apprentices and suggests that an apprentice’s level of social identification with work groups shapes how bullying is identified, interpreted, and prevented. Conclusion: Findings from this study will be important for tailoring evidence-based interventions, human resource policies, and initiatives for education and awareness training. Themes also highlight systemic inadequacies impacting apprentices’ mental health and skill development, with implications for the future sustainability of apprenticeship training agreements.

## 1. Introduction

The psychological and physical abuse of apprentices and trainees is a complex and pervasive phenomenon that has become entrenched in the culture of Australian workplaces [1]. Once non-existent in the scientific literature, workplace bullying has emerged in the last three decades as a well-established topic in psychological, organisational, and legislative discourse [2]. Despite the increasing awareness of workplace bullying, there is a notable lack of literature investigating vulnerable high-risk cohorts such as construction industry apprentices [1,3].

Definitions of workplace bullying feature two core elements: repeated and persistent negative behaviours by the perpetrator and a perceived imbalance of power between the perpetrator and the victim [4]. Einarsen and colleagues [5] have defined workplace bullying as:
“Harassing, offending, socially excluding someone, or negatively affecting someone’s work tasks. In order for the label bullying (or mobbing) to be applied to a particular activity, interaction, or process, it has to occur repeatedly and regularly (e.g., weekly) and over a period of time (e.g., about six months). Bullying is an escalating process in the course of which the person confronted ends up in an inferior position and becomes the target of systematic negative social acts. A conflict cannot be called bullying if the incident is an isolated event or if two parties of approximately equal ‘strength’ are in conflict”.(p. 26)

Workplace bullying has also become a major legal issue for employers, and legal definitions in most jurisdictions vary depending on the context and the nature of the behaviour under investigation [2]. Under Australian workplace law, workplace bullying occurs when “an individual or group of individuals behaves unreasonably towards a worker or a group of workers at work, and that behaviour constitutes a risk to health and safety [6].”

### 1.1. Risk Factors

The emerging literature has demonstrated that systematic workplace bullying is a common experience for apprentices [1,7,8,9]. Trade apprentices experience a range of complex psychosocial risk factors such as emotional dysregulation, relationship problems, psychological distress, substance use, non-completion of apprenticeships, and suicidal ideation [7,10]. A recent study found that 30% of apprentices who experienced bullying and 20% who endured severe bullying reported low financial security, work stress, a lack of support for training and education, and stigma towards mental health [8]. In addition, a cross-sectional study by Pidd et al. [10] found apprentices exposed to routine workplace bullying were at higher risk of poorer mental health outcomes and substance-related harm compared to age- and gender-equivalent population norms. Further complicating these challenges, apprentices do not feel comfortable asking for time off work, do not disclose their problems to their employer [11], and experience difficulty transitioning into the workforce [12]. Specifically, new apprentices experience an abrupt shift into distinct male-dominated workplaces where rites-of-passage induct them into hierarchical structures that emphasise western masculinism and a ‘shut-up and cop it’ culture [1,9,10]. Apprentices and bystanders are also reluctant to report instances of workplace bullying for fear of retribution or having their employment terminated [1,8]. These factors help explain the current high attrition rates of apprentices, where approximately half of trade apprentices are not completing their training agreements [12,13]. Furthermore, unpleasant working conditions, low wages, and poor training have contributed to the long-term skills shortages in trade-based vocations, potentially weakening the Australian apprenticeship system [13].

### 1.2. Suicide and the Workplace Bullying of Apprentices

Associations between workplace bullying and suicide were observed in a large-scale Scandinavian study of nine different occupational groups (*N* = 98,330) by Conway et al. [14]. The study found statistically significant rates of death by suicide, suicidal attempts, and suicidal behaviour in men who had also experienced workplace bullying. Similar associations were observed in Australian blue-collar industries [7,15,16,17,18]. Between 2001 and 2019, the age-standardised suicide rate for male Australian construction workers was 26.0 per 100,000 compared to 13.2 per 100,000 for male workers in other occupations [19]. In 2007, Heller and colleagues [15] were the first to reveal the disproportionately high suicide rates in young construction workers, with male workers aged 15–24 years twice as likely to die by suicide than state and national age-matched cohorts. Focus groups conducted as part of the study suggested that the high suicide rates were associated with workplace bullying. More recently, apprentices’ high exposure to suicide was observed in a large-scale study [8], where approximately one-third of apprentices reported suicidal ideation and 64% reported knowing someone who had attempted suicide in the previous year. Furthermore, the same study found apprentices who experienced suicidal ideation in the previous 12 months were more likely to also report bullying in the workplace [8]. This research highlights that trade apprentices and blue-collar workers in general are among the highest occupational risk categories for suicide. To the best of our knowledge, in the context of the Australian construction industry, no causal links have been established between workplace bullying and suicide; however, Conway et al. [14] suggest that causal mechanisms (i.e., mediating and moderating factors) between workplace bullying and suicidality are plausible given that exposure to workplace bullying has a role in mental health conditions such as depression and suicidal ideation, which are also antecedents to suicidal behaviour.

### 1.3. Social Identity Theory and Workplace Bullying

Social identity theory (SIT) has been widely used by researchers [20,21,22,23] to emphasise important psychological group processes and suggests that people incorporate important group memberships into their self-concepts. According to Haslam et al. [24], groups also provide meaning, purpose, and stability to an individual; thus, group membership or a sense of belongingness can provide a buffer against threats to the individual’s wellbeing. SIT also provides theoretical insights into the formation and maintenance of workplace bullying behaviours and the bullying of trade apprentices.

Escartín et al. [25] assert that cognitive evaluations between members and groups are mediated by self-categorisation processes such as prototype formation and depersonalisation. Group prototypes, therefore, are blurred representations of exemplary or idealistic qualities of members that embody the salient features of group characteristics, represent an imprecise demarcation of group boundaries, and model the behaviours expected from members, distinguishing the high-status in-group and the low-status out-group. Such prototypical characteristics are evident in the apprenticeship bullying literature. The highly male-dominated construction industry is shaped by entrenched social norms that emphasise self-reliance, tolerance of conflict, stoicism, and suppressive emotional regulation strategies [1,14]. According to Ramsay et al. [26], prototypical characteristics ascribed to individuals become stereotypes, which may lead to prototype-based depersonalisation of an individual whose identity does not accord with the in-group [26]. The stereotypical apprentice is described as a “whinger” [1] (p. 414) and “at the bottom of the food chain” [8] (p. 4) due to characteristics such as age, introverted personality, physical characteristics, and their inability to defend themselves against more powerful entities [9]. Such prototypical representations of tradespeople and apprentices have also been observed in the UK hospitality industry [27] and in the US, where medical students are exposed to a training culture of public humiliation and abuse [28].

A controversial theme emerging in the apprenticeship bullying literature is the distinction between bullying and banter [1,8,9]. According to Riggall and colleagues [9], the construction industry is characterised by an informal atmosphere where banter and practical jokes are integral to the culture; however, the authors also suggest that banter is sometimes used by perpetrators to mask bullying because it is directed towards apprentices, usually with intent to harm. However, Alexander and colleagues [29] challenge the notion that all forms of bullying and banter are destructive, arguing that certain bullying-like forms of communication are necessary to allow teams to function at the highest level. Thus, engaging in such behaviour provides symbolic reassurance that the team will not be let down under pressure, which in turn strengthens cohesion and within-group identification. Controversies such as these highlight the ambiguity of bullying tactics, reinforcing that the distinction between bullying and banter is dependent on individual ethics and organisational context, warranting further discussion [1,9,30].

We explore the impact of social identification (i.e., a sense of group membership or belonging) on the bullying of construction industry apprentices and their subsequent reluctance to report such treatment. This damaging and pervasive cycle of abuse in Australian workplaces highlights the need to shift the focus from basic awareness to meaningfully changing the entrenched culture of bullying among trade apprentices. Given that this study builds upon previous qualitative and quantitative research conducted specifically with apprentices [7,8], this study explored the valued attitudes and opinions of a diverse group of skilled and experienced industry representatives who possess a deep and nuanced understanding of the culture of the construction industry and the apprenticeship system in Australia. Therefore, from the perspective of tradespeople, managers, and supervisors in the construction industry, the main aims of this study were to explore how social identification impacts the likelihood of bullying for perpetrators and targets and how social identification impacts the willingness of targets and bystanders to report bullying.

## 2. Materials and Methods

### 2.1. Participants

In order to gain a richer and more holistic understanding of the phenomenon of apprenticeship bullying, this study used maximum variation sampling (MVS) to explore the widest range of perspectives, skills, and experiences possible. Rather than using a homogenous sampling strategy, MVS allowed us to maximise these differences in order to identify and analyse patterns of typical and extreme perspectives across the workplace group [31]. The inclusion criteria were that participants had completed a trade apprenticeship or had suitable trade, supervisory, or professional experience in the construction industry. See Table 1 for the characteristics of the sample.

Mates in Construction (MIC), a construction industry suicide prevention, intervention, and postvention organisation, recruited an initial pool of 15 participants. Contact details were provided to P.G., who then arranged and conducted the interviews. Five participants recruited through MIC did not know the researcher; however, due to participant availability and to control for potential self-selection bias, a further three participants from P.G.’s own industry network were included in the sample. The final sample size of eight participants was sufficient to achieve data saturation, where no new information is obtained and further coding is no longer feasible [32]. The participants were tradespeople, supervisors, and managers who ranged in age from 33 to 72 years (M = 46.4 years), and their time in industry ranged from 10 to 53 years (M = 24.6 years). No female personnel were available to be interviewed, and no other sources of data were used.

### 2.2. Materials

Reflexive thematic analysis (TA) was considered appropriate to explore the aims of this study. Reflexive TA is a flexible method that allows the researcher to identify thematic similarities and differences across the dataset [33]. Given that reflexive TA is not anchored to fixed epistemological assumptions, the research epistemology for this study is situated broadly within a critical realist paradigm, which assumes that reality or events (i.e., instances of workplace bullying) are “always mediated through the filter of human experience and interpretation” [34] (p. 183). Reflexive TA acknowledges the researcher’s own experience and perspectives as an essential resource when interpreting data [35]; thus, an active role in theme generation was adopted using a “bottom-up” approach to code semantic content rather than using a pre-existing coding frame [33]. The author designed and conducted semi-structured interviews with a total of ten open-ended questions. The interview protocol was shaped and guided by SIT and existing bullying literature, using language appropriate to the industry. For example, to explore how participants understood the extent to which targets’, perpetrators’, or bystanders’ level of within-group identification may impact the likelihood of bullying, participants were asked, “How do you think being part of the team or being ‘one of the boys’ impacts an apprentice’s chances of being bullied?” [26]. Similarly, to explore how participants understood inherent power differences within the construction industry hierarchy, they were asked, “In your experience, how do you think apprentices are perceived by tradesmen in terms of the ‘pecking order’ in the construction industry?” [9].

### 2.3. Procedure

As per human research ethics requirements, participants were provided with an information sheet, and all provided written consent prior to participation. Prior to data collection, participants were informed that their participation is voluntary and confidential, and they may withdraw from the study at any time. Since the sample may have included targets, perpetrators, and bystanders of workplace bullying, interviews were conducted in private, and participants were not compelled to answer any questions they did not wish to answer. Participants were advised that if they became distressed as a result of the interview, the interviewer would assist them in connecting to professional help (e.g., Mates or another helpline as per the information sheet).

At the participants’ request, questions were provided ahead of the interview; however, participants were not provided definitions of bullying or harassment by the researcher, instead conveying their own understandings of the phenomenon via the interviews. Each interview ranged from 27 to 54 min in duration (M = 37.5 min) and was conducted between August and September 2022. Three interviews were conducted via telephone, one in-person, and four via videoconference. The interviews adopted a facilitative approach where the participants were encouraged to speak at length about their attitudes and opinions without interruption or time limits. Interviews were audio recorded, and data was de-identified. Otter.ai Pro version(Mountain Creek, NJ, USA) speech-to-text software was used for data transcription, and interviews were checked for accuracy prior to analysis. All audio recordings were erased after transcription. The study was approved by the Griffith University Human Research Ethics Committee (GU reference number 2022/400).

### 2.4. Data Analysis

Selected themes were based on salience and relevance to research aims rather than prevalence in the dataset. The eight transcripts were analysed using NVivo 12 data management software (QSR International, Melbourne, Australia). During phase 1, the transcripts were read twice, and potential codes were highlighted only after deep familiarisation with the interview data was achieved. In phase 2, open coding commenced, where illustrative extracts from the entire dataset were coded and entered into NVivo. Phase 3 involved sorting and combining the codes into broader categories, as well as using annotations and memos entered into NVivo. In phase 4, reviewing themes involved reviewing coded data extracts to determine coherent thematic patterns by combining themes or breaking them down further. Phase 5 consisted of defining and naming themes and consolidating a satisfactory thematic outline. Finally, phase 6 involved producing the report that detailed the analytic narrative in relation to the aims of the research [33]. Final themes were discussed with V.R. until agreement was reached. Additionally, to ensure transparency and trustworthiness of the findings, this study used the Consolidated Criteria for Reporting Qualitative Research (COREQ) checklist [36].

## 3. Results

Four overarching themes were identified: difficulties for apprentices transitioning into industry, the need for continued improvement in industry culture, reluctance to report bullying, and rethinking apprenticeships to empower. See Table 2 for themes and selected illustrative quotes.

### 3.1. Difficulties for Apprentices Transitioning into Industry

Participants identified numerous challenges apprentices must contend with as they transition into the construction industry. The key to this is learning their unique position in the social and organisational hierarchy. It was noted that apprentices always start at the bottom and need to “earn their stripes” to gain credibility as they progress. Participants also highlighted the “culture shock” apprentices experience as they transition from school into industry and how this increases the likelihood of becoming a target of bullying. Those who had previously mentored apprentices emphasised that apprentices coming from a sometimes-sheltered school and home life into a complex and dangerous new work environment are exposed to unknown risk factors. It was emphasised that there is a need to improve apprentices’ communication skills and build resilience to assist them in coping in this new and sometimes hostile environment.

While it was generally acknowledged that bullying is entrenched in the construction industry, the consensus among all participants was that apprentices do not need to be conditioned or “toughened up” through bullying in order to survive the industry. Rather, participants acknowledged that apprentices throughout their four-year training sequence require thorough supervision, mentoring, and education to help them adapt to and thrive in the industry. Several participants who had mentored apprentices over many decades felt strongly that those charged with the responsibility of training needed to be cognisant of apprentices’ unique situations in order to mentor them appropriately.

Participants also described how female apprentices have their own unique challenges when transitioning into the male-dominated blue-collar industries. Participants noted through their own observations that while women still experience bullying, sexual harassment, and unequal treatment due to their physical attributes, they were also able to exhibit their own unique strengths while working in the construction industry. Perhaps indicative of a more inclusive culture, it was acknowledged that there are more females appearing in the industry in various roles with initiatives such as the Women’s National Association of Women in Construction (NAIWIC) and greatly improved facilities for women on construction sites.

### 3.2. The Need for Continued Improvement in Industry Culture

Participants who completed apprenticeships reflected on their own experiences and stated that although the culture of bullying among apprentices has significantly changed over time, there is still more work to do. Several older participants, who undertook their apprenticeships over 30 years ago, described their experiences of initiations or rite-of-passage ceremonies that involved humiliation or physical cruelty (e.g., being tied to a chair or post, being physically hoisted on a winch, or being subjected to painful electric shocks). One participant noted that in the 1990s, depersonalisation of apprentices was commonplace, and his supervisor described him as slightly better than vermin. Other participants recalled the common experience of apprentices being scapegoated to cover incompetent tradespeople for poor-quality work. It was generally agreed that initiation ceremonies have been phased out; however, participants stated traditional rites-of-passage tended to be more physical in contrast to the pervasive psychological abuse that apprentices experience today.

The culture of the construction industry was characterised by participants as dominantly male, where displays of emotion will invite bullying or masked bullying, which they describe as banter. One participant described how any sign of weakness becomes the focus of repeated abuse. The majority of participants agreed that banter is usually intended to be harmless, an icebreaker, or to build camaraderie in order to find common ground between workers. However, whether banter is considered bullying depends on contextual factors, including how well both parties know each other, input from bystanders, and how the banter is received by the target. For participants, the issue of banter seems to highlight the inadequacies of the current system, thus preventing the culture of the construction industry from improving.

Participants expressed support for increased awareness of mental health promotion programmes and initiatives that were not previously part of construction industry culture. It was generally acknowledged that intervention programmes such as Mates in Construction and other mental health services were having a positive impact on the mental health and well-being of all construction workers (not just apprentices). While most participants reported that mental health services are now more visible and taken seriously by businesses and workers, others were less optimistic and maintained the view that workplace programmes are mocked by a small minority and may intensify the poor treatment of apprentices. It was suggested that such programmes may not have the desired effect due to the programmes possibly being “out-of-touch”, and it was recommended that the focus be on tradespeople treating apprentices respectfully rather than apprentices trying to stand up for themselves.

### 3.3. Reluctance to Report Bullying

The problem of reporting bullying and formal escalation procedures was also a dominant theme. All participants agreed that the main roadblock to apprentices reporting bullying was fear of retribution by perpetrators. Specifically, participants believed that if a formal complaint was made against a perpetrator, then this would inevitably lead to more frequent and intense bullying. Other consequences noted by participants included fear of losing employment, labour hire apprentices having their contract terminated prematurely, or apprentices not knowing who to turn to or trust. Participants also described the real social consequences of reporting bullying, such as group ostracisation from the industry, loss of reputation or social credibility, self-perception of failure, and receiving stigmatising labels. Reasons cited for tradespeople and other bystanders not reporting instances of bullying were similar to those of apprentices. In addition, participants described how reporting co-workers was unacceptable team behaviour, with these attitudes giving safe harbour to perpetrators. Problems with reporting policies and how complaints are handled at the organisational level were also identified as contributing to the reluctance to report instances of bullying.

Participants also provided insight into the reluctance to report due to power differences between perpetrators and targets. If an apprentice endured bullying without reporting or complaining, the power dynamic would change eventually, and the apprentice would “earn his stripes.” It was suggested that for targets, some instances of poor treatment were time-limited, with some believing it “was meant to bring out the best in the apprentice in the long term.” Another participant expressed his own feelings of ambivalence towards perpetrators of bullying in his apprenticeship. He recalled his reluctance to report the bullying due to feeling conflicted, helpless, and powerless from the tradespeople who were treating him poorly yet were also charged with the responsibility of mentoring and training him; therefore, he felt that reporting was inappropriate.

### 3.4. Rethinking Apprenticeships to Empower

The final theme identified was the need to reconceptualise how apprentices are engaged in training agreements. Participants expressed the view that apprenticeships are still a pathway to a bright future and that there are many opportunities to travel the world and work in a range of sectors; however, it was noted that mentoring and leadership are pivotal to producing competent and adaptable tradespeople. Participants highlighted that the overuse of labour-hire apprentices is associated with problems within the construction industry such as exploitation, undertraining, and bullying. A key issue stressed by participants was the need for the industry to return to company-indentured apprenticeship programmes and genuine one-on-one supervision. Participants agreed this would allow companies to provide a more focused mentoring relationship, allow for better provision of skills training, and increase apprentices’ sense of identification with the industry.

Factors affecting motivation and performance were also a topic of interest for participants as they considered how apprenticeships might be reconceptualised. Some stated they had witnessed apprentices who were highly esteemed by their companies and thus given more responsibilities than others. Participants stressed that these apprentices were usually mentored by tradespeople because they had shown initiative and were more motivated, as tradespeople were less inclined to invest time in apprentices who were low performers. Participants stated that although supervisors try to use a measured approach with a level of difficulty for job progression, complications arise when apprentices are exposed to physically demanding tasks intended to condition them to the risks and dangers inherent in trade-based occupations. Participants explained that certain tasks apprentices are given under correct supervision will determine to what extent they can be trusted with higher-level tasks, especially in the electrical trades where the risk of fatality is ever-present. They stated that whether being relegated to these physically demanding tasks is considered bullying is a fine line and dependent on tradespeople’s attitudes, the type of trade, and the context in which the task was given.

According to participants, apprenticeship programmes could be improved to have a positive impact on apprentices’ wellbeing and reduce bullying by encouraging greater acceptance of apprentices into the workplace culture. It was suggested that apprentices’ input should be taken on board, and their voice needs to be heard through community forums and social events to build relationships and increase identification with work groups. Participants also emphasised the need for companies to have a stronger endorsement of policies and codes of conduct to outlaw bullying, with clear messaging regarding what is appropriate conduct and a zero-tolerance approach to bullying. Implementation of such conditions of employment under relevant workplace laws will also provide procedural fairness to those under investigation.

## 4. Discussion

The first theme identified was defined by the difficulties faced by apprentices as they transition into the workforce. These results were supported by the first tenet of SIT, where individuals will join groups if membership is likely to maintain or improve their self-concept [20]. Participants indicated that an apprenticeship is a viable career pathway in the construction industry. However, for young people to become a part of the construction industry, it also means integrating into their self-concept social norms such as bullying, stoicism, invulnerability, and the avoidance of negative emotions [16,37]. While the results of this study suggest identification with industry impacts female workers and non-completion rates, participants also explained there is a relationship between factors that increase inter-group differentiation and subsequent workplace bullying (i.e., apprentice suitability and labour hire stigma). The results of the present study suggest support for Ramsay et al. [26], who proposed that social categorisation relies on salient characteristics that form in-group identification, and Escartín et al. [25], who found that higher social identification reduced the likelihood of bullying from in-groups and out-groups. Similarly, the effect of increased social identification with an in-group may in turn reduce the depersonalisation of apprentices and the subsequent likelihood of bullying [26]. Thus, the extent to which an apprentice’s identity is integrated into healthy work groups and mentoring relationships in order to develop resilience strategies and communication skills, the more likely apprentices will be able to appropriately interpret and deal with instances of workplace bullying.

Participants reported that the culture of the construction industry has shifted in the last two decades from a culture of ritual initiations into an awareness of more egalitarian social norms. The emerging culture includes increased awareness of mental health and suicide awareness programmes, in line with previous research [7,11]. This cultural shift also suggests an alignment with an increased understanding and awareness of the relationship between masculine norms and psychopathology and the conditions under which it occurs [37]. The culture change noted by participants is consistent with research by Turner and Reynolds [38], who found that overt acts of bullying have become more subtle, resulting in differing social motivations between targets and perpetrators. While low-status apprentices seek favour and increasing levels of social support from the high-status group as they become tradespeople [26], the bullying of weaker apprentices can be characterised by out-group mistrust and depersonalisation through stereotyping, especially by groups that hold negative social rules [26]. However, the eradication of old-school initiation rituals, as stated by participants, may have been exchanged for more subtle types of negative communication, such as banter, with previous research suggesting this may be a reaction to more stringent workplace legislation and policies [39]. Participants in this study suggested banter may have a dual effect on pro-social outcomes such as bonding and higher in-group identification [27,29]. However, banter may also mask workplace bullying by exploiting existing power imbalances between perpetrators and targets. The results of the present study are consistent with the previous literature that says whether banter is considered bullying depends on the relationship between the perpetrator and the target, the role of the bystander, the perpetrator’s intention, and the interpretation of the behaviour by the target [1,26,29].

The widespread culture of non-reporting of bullying is consistent with current apprenticeship bullying literature [1,8,9,27]. However, despite apprentices’ need to maintain viable working relationships with tradespeople, the fear of retaliation [1,8] and a sense of traditional stoic ‘stiff upper lip’ [40] were reported by participants. Long-standing research into group dynamics holds that low-status groups will tend to favour the high-status group to the extent that they identify with the self-concepts of its members, view intergroup boundaries as stable, and view the intergroup relationship as legitimate [38]. The results of the present study suggest that targets are unwilling to report bullying as they still perceive perpetrators of the high-status group as legitimate, despite the acknowledgement that continued exposure to workplace bullying has significant adverse effects on the mental health and wellbeing of apprentices [1,8,9]. This is especially the case considering that bullying targets are forced to rely on perpetrators of bullying for their trade training, placing themselves in an onerous position. Therefore, given the assertions made by the participants, it is conceivable that any formal reporting of workplace bullying may be perceived by perpetrators as a symbolic rejection of the legitimacy of the status quo through “anti-norm deviance” [41] (p. 911) and an affront to the existing hierarchy, causing a cycle of further retaliatory bullying. The results of the present study also found bystanders who supported the perpetrator or were more likely to fear retaliation from the perpetrator, which was consistent with the apprenticeship bullying literature [1,8,9].

The final theme identified how meaningful change is required to empower apprentices and modernise the sustainability of the apprenticeship system. This theme acknowledges that change is required in order to address the systemic and cultural issues that increase the frequency and intensity of bullying, improve training outcomes, and increase the general wellbeing of apprentices. Consistent with Snell and Hart [13], the results suggest there is an urgent need to decrease reliance on apprentices recruited through group apprenticeship schemes. The participants explained that in light of current skills shortages and in order to attract financial incentives, companies will increase the uptake of apprentices at the expense of reduced skills and competencies. This has also caused apprentices engaged in these schemes to be stigmatised as “labour hire apprentices.” These calls for systemic change are supported by a recent national review of apprenticeship programmes by Stanwick and colleagues [42], who recommended financial incentives to employers need to be carefully gauged to prevent inferior outcomes and unintended consequences for apprentices. Moreover, the sentiments of participants provide further support for Stanwick et al. [42], where completion rates, issues with on-the-job training, and the relevance of the current approach in a twenty-first century context were considered “pain points” [42] (p. 1) and compared unfavourably to approaches by other countries. Recommendations from Stanwick et al. [42] included shortening the length of the apprenticeship, “front-loading” [42] (p. 18) training before employment commences, and establishing best practice training to increase employers’ return on investment. A viable solution, according to the participants of the present study, is to return to company-indentured apprenticeships where an apprentice is employed exclusively by one employer for the duration of the apprenticeship, thus reducing the stigma and over-reliance of group apprenticeship schemes. Consistent with previous research [24], increased social identification with work groups is shown to improve an individual’s long-term health, well-being, and morale; thus, apprentices can be empowered with a sense of purpose whereby they can be integrated into a company structure that emphasises the role of mentoring and training relationships.

## 5. Limitations

Several potential limitations of this study should be considered in terms of the results. First, the different interview modalities that were used (i.e., video conference, telephone, and in-person) may have impacted the rapport with participants. However, given that previous analyses of qualitative interviews found only minimal differences between modes of providing rich and sensitive information, it is unlikely that these differences impacted these results [43,44]. Second, given the ontological underpinnings of the study, the sample size of eight, and the fact that the final themes were guided by the salience and relevance of the aims of this study, the results do not necessarily generalise to the experiences of other individuals or groups working in the construction industry. As such, further studies could include a quantitative representation of diverse groups to gain a better understanding of how group identity impacts discrimination, mental health, and suicidality in the construction industry [13]. Finally, given the emotional and volatile nature of the topic, some participants may have used the interview as an opportunity to provide socially desirable anti-bullying responses. Thus, the interpretation of these results may be impacted by self-selection bias due to participants who were motivated to “make a statement.” Future research may use screening procedures to control for potential sources of bias.

## 6. Conclusions

The results of this study are consistent with previous research and offer novel insights into the phenomenon of workplace bullying within the context of the Australian building and construction industry. This study used SIT as a theoretical framework to explore the impact of social identification on targets, bystanders, and perpetrators from the perspective of leaders in the construction industry. The results of this study may be used to inform the development of evidence-based interventions as well as human resource policies and initiatives for education and awareness training to reduce the impacts of bullying among trade apprentices. These initiatives will be pivotal in providing appropriate levels of support to improve the mental health and well-being of apprentices so that targets and bystanders become sensitised to bullying tactics. Moreover, given the prevalence of apprenticeship bullying and dedicated support from the construction industry, it is hoped the results of this study will further reinforce the need to facilitate cultural change among all stakeholders that workplace bullying is no longer acceptable in Australian workplaces.

## Figures and Tables

**Table 1 ijerph-20-06980-t001:** Sociodemographic characteristics of the sample.

Characteristic	Frequency (*n*)
Current employment	
Tradesperson	2
Supervisor	2
Management	3
Left the industry	1
Highest qualification	
Trade	1
Advanced trade	1
Diploma	2
Bachelor’s degree	4

*N* = 8. All participants were English-speaking males.

**Table 2 ijerph-20-06980-t002:** Final themes with codes and illustrative quotes.

Theme	Code	Illustrative Quotes	Frequency (*n*)
Difficulties faced by apprentices transitioning into the industry	Attitudes towards toughening up	“Definitely not! [Apprentices] definitely do not need to be toughened up. Industry needs to acknowledge that through education and supervision… There is no toughening up.” (Participant 2)	10
	Gaining Respect	“I think that apprentices need to earn the right to be in a certain social group or have a certain amount of responsibility or be treated with a certain amount of respect.” (Participant 1)	9
	Non-completion	“I would say, you know, that’d be a good chunk of why people [apprentices] drop out because they are just like, no [expletive] this I did not sign up for this [bullying]. You end up not wanting to go work at all.” (Participant 8)	5
	Transitioning into industry	“We are dealing with apprentices that have come off the couch looking at Instagram. Suddenly on a construction site surrounded by cranes, other guys, unions, this whole new world starting work at six o’clock in the morning and some of them are just glassy eyed and I do not blame them.” (Participant 6)	9
The need for continued improvement in industry culture	Old-school comparisons	“I think if we go back, say another… Let’s say we went back another 15 [years]. I would hate to think what it was like then, but I’d say we’ve probably in the last 15 come a long way.” (Participant 3)	14
	Initiation ceremonies	“It did have the initiations that were sort of instigated by senior apprentices, but apart from that I did not agree with it when I was in my first year... so as time went on, I suppose that got diminished and got phased out.” (Participant 7)	5
	Depersonalisation	“Do not think that you are anything special. He said the only things that you are higher than in this [workplace] are the cockroaches and that is only by about that much." (Participant 1)	10
	Mental health awareness	“I mean there’s a lot more information out there now with you know, through Mates and Are You OK days so there’s a big there’s a big presence now of people trying to have that conversation and it is definitely something that I can see the improvement in last two years with our company.” (Participant 6)	3
Reluctance to report bullying	Retaliation and repercussions	“Yeah, look I think there’s a variety of factors… It [non-reporting] could be [fear of] further bullying, it could be worse bullying, it could be fear of losing a job, it could be fear of… maybe out-casting themselves further, getting bad names themselves in the industry.” (Participant 5)	5
	Tradespeople are not reporting	“I possibly should have stepped up and reported it myself. But again, I look back at that and I go well, it was… it was just not the done-thing.” (Participant 1)	9
	How cases are handled	“If they do report it and if no action is taken then it just represents a poor culture. Then then they [apprentices] get to the point, what is the point of telling them that is the other thing we need to mindful of.” (Participant 6)	9
	Sticking it out	“Some apprentices think that it is not what you do, you just take it [bullying]. There’s a time limit on that, I will not be an apprentice forever, eventually I will be qualified. You know, the power imbalance will not be there anymore. So, some might just stick it out… stiff upper lip, just get through it.” (Participant 1)	2
Rethinking apprenticeships to empower	Company policies	“Education I think is very important. So that is for all employees in the business. You have the code of conduct training, your bullying, harassment, and discrimination training. So, people know where’s the line in the sand and what is appropriate, what is not appropriate.” (Participant 4)	9
	Support networks	“You need to look after the apprentices… like your son or daughter. You got to look after them. It comes through education and the person training, that is the number one step, someone needs to guide them through their four years, someone who truly cares.” (Participant 2)	12
	Calls for mentoring	“You’ve gotta have structure within the organisation… but separate to that direct supervisor should have a mentor… either a 2-up mentor that might be a boss of the supervisory level [and] that you that you’d have regular catch ups.” (Participant 4)	15
	Showing empathy	“Bringing them [apprentices] into the fold, including them in the training not disregarding their input… So, if the apprentices do have a voice, that voice needs to be heard, not just ignored… I think it is big plus.” (Participant 6)	4

*N* = 8. Frequency refers to the number of times the code appears in the dataset.

## Data Availability

The data presented in this study are available on request from the corresponding author, V.R. The qualitative data are not publicly available due to their confidential signature.

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
