# Peer review of "Exploring the Impact of Social Identity on the Bullying of Construction Industry Apprentices"

_ijerph, 2023, doi:10.3390/ijerph20216980_

Round 1
Reviewer 1 Report
Comments and Suggestions for Authors
Dear Colleagues,
in my opinion the paper is for sure of interest since the topic is timely and relatively underresearched. Additionally, the focus of the study is appropriate for IJERPH. Despite these good points, however, I have a number of concerns regarding the manuscript, that I detail here below.
1. I miss a clear definition of bullying in the first part of the Introduction. This would provide more clarity to the paper. I would focus on the definition provided in the international book: Bullying and Harassment in the Workplace by Einarsen et al. (2020).
2. P. 2. Section on bullying and suicide. You may also highlight that the bullying-suicide link has been established in different studies – please see this recent study by Conway et al (2022) - doi:10.5271/sjweh.4034
3. You may generalize a bit your Introduction by considering that bullying of young workers is not limited to the construction industry - think to doctors, for example. There should be some literature to rely upon. See, for example, https://www.ncbi.nlm.nih.gov/pmc/articles/PMC101400/.
4. P. 3 (Participants). What is “maximum variation sampling”. You should provide more explanation about this sampling strategy.
5. On P. 3 it is reported that: “Inclusion criteria were that participants had completed an apprenticeship in a blue-collar industry or had suitable trade, supervisory or professional experience in blue-collar or related industries”. I think this is at odds with what you report in the title and the Introduction of the paper that the focus will be “Construction Industry”. Can you please clarify?
6. On P. 4 (beginning) it is reported that “the final sample size … was sufficient to ensure data saturation”. I have concern with a sample of only eight participants, since the potential for generalization is really limited. This should be discussed in the limitation section of the paper more in depth. Additionally, you should also elaborate more on the statement that a sample of eight was enough for data saturation. What does it mean exactly? To my knwoledge, a sample of eight is to small.
7. Relatedly, apparently the sample doesn’t include apprentices, or at least this is not clear. If this is true I think the study has an important weakness, in the sense that the aim is to talk about apprentices experiences but the voice of apprentices (of the construction industry) is not represented. Can you please clarify?
8. In the end of the Material section it is reported that: “The author designed and conducted semi-structured interviews with a total of ten open-ended questions. The interview protocol was shaped and guided by SIT and existing literature”. I think you should elaborate more on this, including reporting example (or even all) of the questions used in the protocol.
Author Response
Word document 'Responses to Reviewer 1' is attached.

Reviewer 2 Report
Comments and Suggestions for Authors
a very interesting paper
I would have stresses maybe more the link of these practices towards the apprentice with workforce shortage and lack of attractivity of the construction sector ( page2)
maybe a short highlight on the legal context in Australia as it is mentionned in the discussion (page 9) would be useful in the first part of the paper
another suggestion : to show more clearly the link between the 10 open ended questions ( we don't know about the questions ?) and the main elements of SIT and litterature supporting the interview guide
In future research to target a sample of apprentices as the actual sample is "older" such as tradespeople, supervisors and managers -age sample 33 to 72.
The main question addressed by the research is Workplace bullying among construction apprentice.
The topic is original and relevant to the field and it address a specific gap in the field. There is a growing interest in the topics, linked with Managerial practices based on masculine and sometimes sexist values. Same topics are focused on in hospitality industry Another sector where workforce shortage is a growing issue.
It add the vision of managers and older people / drawing on their perceptions to the subject area compared with other published material.
Maybe taking into account the fact that the sample is relying on memories / on of my demand was what are the questions of the interview Guide.
The conclusions seems so / and consistant with some research on hospitality in France among star class restaurants work force.
The references are appropriate - seems the paper is correctly documented.
Author Response
Word document "Responses to Reviewer 2' is attached.

Reviewer 3 Report
Comments and Suggestions for Authors
The topic discussed in the article is really interesting and not sufficiently studied before. The approach of the authors is acceptable. However, the studied sample of respondents seems too small to draw significant conclusions. I suggest the authors expand a sample of this study and resubmit the article with more reliable results and findings.
Author Response
Word document "Responses to Reviewer 3' is attached.

Reviewer 4 Report
Comments and Suggestions for Authors
Thank you for giving me the opportunity to review this manuscript.
From the authors: “Using social identity theory (SIT), the aim of this study was to gather the attitudes, thoughts, and feelings of construction industry leaders to better understand how social identification (i.e., group membership) impacts bullying on targets and perpetrators, and the willingness to report bullying for targets and bystanders.”
As per the literature review, this investigation looks into an area which has been mostly neglected in the workplace bullying literature, particularly in Australia, young apprentices in the building industry. As such, it has an important contribution to make.
Some of my concerns which informed my suggestion to the editor are as follow:
Unfortunately, in the literature review there is no definition of bullying provided, particularly bullying in the workplace. This causes some confusion for example when confounding bullying and banter which are not the same thing. Intent to harm, does not make banter bullying. Banter, as in teasing, intends to cause some harm in order to alert the receiver that they may be violating a social norm. The intent is to teach and include the receiver if they conform to the rule and the banter reduces at the rate that they conform. Bullying and banter therefore have diametrically opposed outcomes and as such you could hardly argue that some forms of bullying increase performance or behaviours are perceived as bullying or not purely on the cultural ethic. The outcome to the individual mediates its perception of bullying or banter. Alexander et al, whom they cite, concluded thus.
I am also confused by the need to gender certain behaviours as being typical of one group or another. And why is there a need to negatively label these behaviours? For example, can't women be stoic, self-reliant? Why is not talking about your emotions necessarily suppression? The implication is that these traits are (a) masculine and (b) necessarily negative. If that is the case how do the authors explain the fact that across most mental health problems women are overrepresented? Including suicide attempts at four times the rates of men? Is the purpose of this manuscript to investigate bullying or to comment on evaluative assumptions about stereotypical behaviours in men and women? My point is that the behaviours described are not inherent to male workplaces. The authors themselves quote studies that have been conducted with female hairdresser apprentices in which these behaviours were found. The literature review by Salin (2021; Workplace bullying and gender: An overview of empirical findings Dignity and inclusion at work, 331-361) for example showed that rates of bullying were higher for women dominated workplaces, however, conceding that this was to do with complex interactions of gender expectations in which both females and males were more likely to be bullied if they did not meet those norms by gender matched peers.
When I reviewed the references provided for the following statement for example “The construction industry is shaped by rule-based norms that emphasise masculine traits such as self-reliance, tolerance of conflict, stoicism, and suppressive emotional regulation strategies” (p2 L87-90) neither of them supported the gendering of the behaviours as described by the authors or their conclusions.
There are other areas in which this manuscript genders all negative behaviours in the construction workplace as male and negative. If the purpose of this manuscript is to engage in gender analysis of bullying behaviours, I do not believe that this is the correct journal for this paper, and perhaps another avenue could be suggested. Its inclusion in the current manuscript detracts from the possible contribution the manuscript has to offer.
In respect to the methodology, The methodology used is from a qualitative philosophical/ontological stance with a total sample size of 8. The researchers would be aware that from a methodological perspective, their methodology does not lend itself to making generalisations to the rest of the population as would a positivist strategy. The researchers note that “TA is not anchored to fixed epistemological assumptions, the research epistemology for this study is situated broadly within a critical realist paradigm, which assumes that reality or events (i.e., instances of workplace bullying) are “always mediated through the filter of human experience and interpretation”” p4,L142-145. The paper is written however, as if it were a positivist study, particularly in relation to the recommendations and implications of the study. The implications need to be re-worked consistent with the ontological approach of the methodology and the authors professed non-positivistic stance. Readers will be misguided by considering that the conclusions and recommendations of this study are based on a representative sample that can be generalised to the rest of the population. The study represents the lived in experience of the 8 participants, only. Due to their own world experience this is their personal experience and as individuals their personal experience is not transferrable to others. Further, the analysis was guided by the aims, as such it was confirmatory rather than exploratory, “Selected themes were based on salience and relevance to research aims rather than prevalence in the dataset” p4, L172-173, which is fine, however it further adds to lack of generalisability of the results as the results do not represent most frequent statements for example but statement consistent with the aims of the study. These limitations are not addressed by the statement on limitations in the paper. There should not be an attempt to generalise if the study is to be consistent with its philosophical and methodological underpinnings.
I would suggest the authors access the work of Braun and Clarke on the use of TA when formulating their revision (Virginia Braun & Victoria Clarke (2023) Is thematic analysis used well in health psychology? A critical review of published research, with recommendations for quality practice and reporting, Health Psychology Review, DOI: 10.1080/17437199.2022.2161594). I note that their 2006 paper was quoted.
Based on these conserns I find it difficult to recommend the papaer for publication in its current form.
Author Response
Word document 'Response to Reviewer 4' is attached.

Reviewer 5 Report
Comments and Suggestions for Authors
Line 47 apprentices. Is repeated better to replace by eg pronoun
There is talk of harassment, with reference to jokes, its relationship with suicide, but there is no definition of what harassment really is. It would be convenient to include it
Before beginning the interviews, it was explained to the participants what it was or was not harassment.
Line 214 y ss It was noted that females still experience bullying, sexual harassment, and unequal treatment . What sector are you referring to? No women participated in this study. It is not clear.
I note that the sample is very small and carried out in a very short period of time. It would be good to widen the sample and the variety of participants
Comments on the Quality of English LanguageThe language is clear, I would review the writing to improve understanding of the text.
Author Response
Word document 'Responses to Reviewer 5' is attached.

Round 2
Reviewer 1 Report
Comments and Suggestions for Authors
Dear authors,
Thank you for taking care of my comments. The manuscript improved. I'm almost ok with your responses. An exception is your description of maximum variation sampling (see my previous comment 4 and your response). Regarding this, I ask you to provide a more articulated description of this sampling scheme since the amendments made are still not enough to clarify the peculiarities of this strategy. Additionally, in my comment 6 I suggested to acknowledge in the Limitation section that the low number of participants included may be a threat to the generalizability of the findings. This has not been explicitly mentioned in the manuscript. Therefore, I suggest you to be more explicit on this when you discuss the limitations of the study.
Best regards.
Author Response
Reviewer 1
Dear authors,
Thank you for taking care of my comments. The manuscript improved. I'm almost ok with your responses. An exception is your description of maximum variation sampling (see my previous comment 4 and your response). Regarding this, I ask you to provide a more articulated description of this sampling scheme since the amendments made are still not enough to clarify the peculiarities of this strategy. Additionally, in my comment 6 I suggested to acknowledge in the Limitation section that the low number of participants included may be a threat to the generalizability of the findings. This has not been explicitly mentioned in the manuscript. Therefore, I suggest you to be more explicit on this when you discuss the limitations of the study.
Best regards.
We thank the reviewer for their feedback, and we offer the following responses to their questions:
- “An exception is your description of maximum variation sampling (see my previous comment 4 and your response). Regarding this, I ask you to provide a more articulated description of this sampling scheme since the amendments made are still not enough to clarify the peculiarities of this strategy”.
Thank you for highlighting this, we agree it is important to provide a better description of this sampling technique. In the manuscript, I have provided a clear description of maximum variation sampling and how it benefits our study. We hope the additional clarification is satisfactory to the reviewer.
- “Additionally, in my comment 6, I suggested to acknowledge in the Limitation section that the low number of participants included may be a threat to the generalizability of the findings”.
Thank you to the reviewer for the opportunity to provide additional clarification in relation to generalisability. In the limitations section of the manuscript, I have edited the text to acknowledge the small sample size within the context of the ontological and methodological limitations of our study. To avoid confusion, we would also kindly refer to the reviewer to our original response to point 6 of their feedback in relation to sampling and data-saturation. In order to provide additional clarification to the reviewer, I note the following points:
Given the ontological underpinnings of the study, we would again highlight the distinction between positivist and non-positivist strategies and the reviewers use of the term “generalisability” in qualitative studies. Positivist strategies use a probability based random sampling techniques (or variation thereof) to obtain a representative sample that is generalisable to the population of interest. However, as the reviewer would be aware, the strategy in our study was quite different to a positivist strategy as it used a non-probability approach, thus by definition it is not simply the small sample size alone that prevents the generalisability of qualitative findings. Generally, the goal of qualitative studies is not to generalise but to provide a rich, contextualised understanding of human experience.
In relation to sample size, scholars have investigated sample sizes for adequate data saturation and recommended sample sizes generally range from 6 to 17. We achieved data saturation with 8 participants. This means no new information was obtained, and no new coding was feasible to answer our research aims. The qualitative data gathered was their personal experience and as qualitative researchers we try to get as “close” to the participants story as possible as their personal experience cannot transfer to or from others. Therefore, sample size is largely irrelevant, it is the participants own unique experience that limits generalisability. Notwithstanding, maximum variation sampling does not increase representativeness and generalisability, what it does is it allows greater insight into the phenomenon being investigated by identifying and examining typical and extreme perspectives.
The results of this study should be interpreted in the context of the non-positivist stance under which the study has been undertaken. We agree that readers will be misguided by considering that the conclusions and recommendations of this study are based on a representative sample that can be generalised to the rest of the population.
References
Cresswell, J. (2016). Qualitative Inquiry & Research Design: Choosing among five approaches. Sage Publications.
Ando, H., Cousins, R., & Young, C. (2014). Achieving saturation in thematic analysis: Development and refinement of a codebook. Comprehensive Psychology, 3, 03-CP.
Gill, S. L. (2020). Qualitative sampling methods. Journal of Human Lactation, 36(4), 579-581.
Hennink, M., & Kaiser, B. N. (2022). Sample sizes for saturation in qualitative research: A systematic review of empirical tests. Social science & medicine, 292, 114523.
Reviewer 3 Report
Comments and Suggestions for Authors
The current version of this paper looks better.
Author Response
We thank Reviewer 3 for their feedback
Reviewer 4 Report
Comments and Suggestions for Authors
Thank you for your attention to my the suggestions provided under review.
Author Response
We thank Reviewer 4 for their feedback.